# The endoplasmic reticulum stress-autophagy pathway controls hypothalamic development and energy balance regulation in leptin-deficient neonates

Soyoung Park [1], Aleek Aintablian[1], Berengere Coupe[1,2,3] & Sebastien G. Bouret[2,3✉]

Obesity is associated with the activation of cellular responses, such as endoplasmic reticulum (ER) stress. Here, we show that leptin-deficient *ob/ob* mice display elevated hypothalamic ER stress as early as postnatal day 10, i.e., prior to the development of obesity in this mouse model. Neonatal treatment of *ob/ob* mice with the ER stress-relieving drug tauroursodeoxycholic acid (TUDCA) causes long-term amelioration of body weight, food intake, glucose homeostasis, and pro-opiomelanocortin (POMC) projections. Cells exposed to ER stress often activate autophagy. Accordingly, we report that in vitro induction of ER stress and neonatal leptin deficiency in vivo activate hypothalamic autophagy-related genes. Furthermore, genetic deletion of autophagy in pro-opiomelanocortin neurons of *ob/ob* mice worsens their glucose homeostasis, adiposity, hyperphagia, and POMC neuronal projections, all of which are ameliorated with neonatal TUDCA treatment. Together, our data highlight the importance of early life ER stress-autophagy pathway in influencing hypothalamic circuits and metabolic regulation.

[1] The Saban Research Institute, Developmental Neuroscience Program, Children's Hospital Los Angeles, University of Southern California, Los Angeles, CA 90027, USA. [2] Inserm, Laboratory of Development and Plasticity of the Neuroendocrine Brain, Jean-Pierre Aubert Research Centre, UMR-S 1172, 59000 Lille, France. [3] University of Lille, FHU 1,000 Days for Health, 59000 Lille, France. ✉email: sebastien.bouret@inserm.fr

Childhood obesity is a serious health problem worldwide. The percentage of children and adolescents affected by obesity has more than tripled over the past 40 years, and nearly 20% of children aged 6–18 years have obesity in the United States[1]. Because childhood obesity is associated with many life-threatening risks, including type 2 diabetes, there is an important need to understand the factors and mechanisms involved in the development of these pathological conditions.

Leptin is an adipocyte-derived hormone that has originally been described to suppress appetite through its action on pro-opiomelanocortin (POMC) and agouti-related peptide (AgRP)/neuropeptide Y (NPY) neurons located in the arcuate nucleus of the hypothalamus[2,3]. The effects of leptin are mediated by a distributed neural network that includes neurons located in various parts of the hypothalamus and in the midbrain and hindbrain[4]. Although leptin does not appear to inhibit food intake and body weight during neonatal life[5], rodents display a sharp surge in circulating leptin levels during the first 2 weeks of postnatal life that appears to exert neurotrophic effects on hypothalamic neural circuits[6,7]. The density of projections from ARH neurons to other hypothalamic sites involved in the control of food intake is severely disrupted in postnatal leptin-deficient *ob/ob* mice and remains diminished throughout life[7]. In addition, leptin exerts its neurotrophic effects during a restricted postnatal critical period: treatment of adult *ob/ob* mice with leptin did not restore ARH projections, but daily injections of leptin during neonatal life and up to peripubertal age rescued these structural alterations[8].

A variety of physiological and pathological situations produce alterations in the endoplasmic reticulum, a condition known as endoplasmic reticulum (ER) stress. ER stress activates a complex intracellular signal transduction pathway called the unfolded protein response (UPR). The UPR is tailored essentially to reestablish ER homeostasis. Previous studies have demonstrated that ER stress and UPR signaling pathway activation play important roles in the development of obesity-induced insulin resistance and type 2 diabetes[9]. Moreover, genetic loss of the unfolded protein response transcription factor spliced X-box binding protein 1 (Xbp1s) causes leptin resistance and promotes weight gain on a high-fat diet. In contrast, the induction of Xbp1s in POMC neurons protects against diet-induced obesity and improves leptin and insulin sensitivity[10]. Furthermore, reversal of ER stress with chemical chaperones, i.e., agents that have the ability to increase ER folding machinery, increases insulin sensitivity and reverses type 2 diabetes in adult *ob/ob* mice and improves leptin sensitivity in adult obese mice fed a high-fat diet[11,12]. However, whether changes in the perinatal environment, such as neonatal leptin deficiency, cause ER stress and whether neonatal ER stress contributes to long-term metabolic regulation remains to be investigated.

Here, we show that during early postnatal life and throughout adulthood, leptin deficiency causes elevated ER stress in various metabolically relevant tissues and particularly in the arcuate nucleus of the hypothalamus. We also report that relieving ER stress in *ob/ob* neonates has long-term effects on metabolic regulation and hypothalamic development. Finally, we find that the mechanisms underlying the effects of ER stress on *ob/ob* mice involve autophagy.

## Results

**Leptin deficiency induces early life ER stress**. To examine whether leptin deficiency causes ER stress during critical periods of development, we measured the expression levels of the following ER stress markers: activating transcription factor 4 (*Atf4*), 6 (*Atf6*), X-box binding protein (*Xbp1*), glucose regulated protein GRP78 (referred to as *Bip*), and CCAAT-enhancer-binding

protein homologous protein (*Chop*), in the embryonic, postnatal and adult hypothalamus of leptin-deficient (*ob/ob*) and wild-type (WT) mice. The levels of ER stress gene expression were not significantly changed in the hypothalamus of E14.5 *ob/ob* embryos (Fig. 1a) or in the arcuate nucleus of the hypothalamus (ARH) of postnatal day (P) 0 *ob/ob* mice (Fig. 1b). In contrast, all ER stress markers examined were significantly elevated in the ARH of P10 *ob/ob* mice (Fig. 1c). We next assessed ER stress marker expression specifically in arcuate *Pomc* and *Agrp* neurons and found that the levels of *Atf4*, *Atf6*, *Xbp1*, and *Bip* mRNAs were higher in these two neuronal populations in P10 *ob/ob* mice (Fig. 1d). During adulthood, leptin deficiency only caused an increase in *Xbp1* and *Bip* mRNA expression in the ARH (Fig. 1e). In addition, *Xbp1* mRNA levels were significantly higher in the paraventricular nucleus of the hypothalamus (PVH) of P0 (Fig. 1f) and P10 *ob/ob* mice (Fig. 1g), but none of the ER stress markers studied were significantly elevated in the PVH of adult *ob/ob* mice (Fig. 1h). We also examined ER stress markers in metabolically relevant peripheral tissues and found that *Xbp1* gene expression was upregulated in the liver and adipose tissue of P10 *ob/ob* mice and that *Atf4* mRNA levels were increased in the livers of P10 *ob/ob* mice (Supplementary Fig. 1a, b). In contrast, most of the ER stress markers studied were downregulated in the liver and adipose tissue of adult *ob/ob* mice (Supplementary Fig. 1c, d).

To determine whether the induction of the ER stress observed in *ob/ob* mice was caused by the lack of leptin, we treated *ob/ob* mice with leptin between P4 and P10 and measured ER stress markers in the ARH. Neonatal leptin treatment reduced the levels of *Atf4*, *Atf6*, *Xbp1*, *Bip*, and *Chop* mRNAs in the ARH of P10 *ob/ob* mice (Fig. 1c). We also performed a series of in vitro studies in which we exposed hypothalamic mHypoE-N43/5 cells to leptin and tunicamycin (an agent commonly used to induce ER stress) and measured ER stress markers. As expected, tunicamycin treatment in vitro increased the mRNA expression of the ER stress-related genes *Atf4*, *Atf6*, *Xbp1*, *Bip*, and *Chop* (Fig. 1i), and leptin reduced tunicamycin-induced expression of *Atf4*, *Atf6*, *Xbp1*, and *Bip* mRNAs (Fig. 1i).

**Neonatal TUDCA treatment improves energy balance**. To investigate the functional importance of early life ER stress in *ob/ob* mice, we treated *ob/ob* neonates daily with tauroursodeoxycholic acid (TUDCA) from P4 to P16, which represents a critical period for hypothalamic development[7]. TUDCA is a chemical chaperone of low molecular weight that has been shown to increase ER function and decrease the accumulation and aggregation of misfolded proteins in the ER lumen and, consequently, relieve ER[12,13]. We first verified that peripheral TUDCA injections reduced hypothalamic ER stress and found that *ob/ob* mice treated with TUDCA neonatally displayed a reduction in ER stress gene expression in the ARH and the PVH at P10 (Fig. 1c, g). A similar improvement of ER stress upon TUDCA treatment was observed in P10 liver and adipose tissue (Supplementary Fig. 1a, b). However, the effect of neonatal TUDCA injections on ER stress gene expression did not persist into adulthood (Fig. 1e, h and Supplementary Fig. 1c, d).

Beginning at 3 weeks of age, the TUDCA-treated *ob/ob* neonates had significantly lower body weights than the control *ob/ob* mice, and this reduction in body weight persisted until 6 weeks of age (Fig. 2a). Neonatal TUDCA treatment also reduced the daily food intake in 10-week-old *ob/ob* mice (Fig. 2b). Moreover, 4-week-old *ob/ob* mice treated with TUDCA neonatally displayed a reduction in fat mass, but this effect was not observed in 10-week-old animals (Fig. 2c). Similarly, respiratory quotient, energy expenditure, and locomotor activity during the

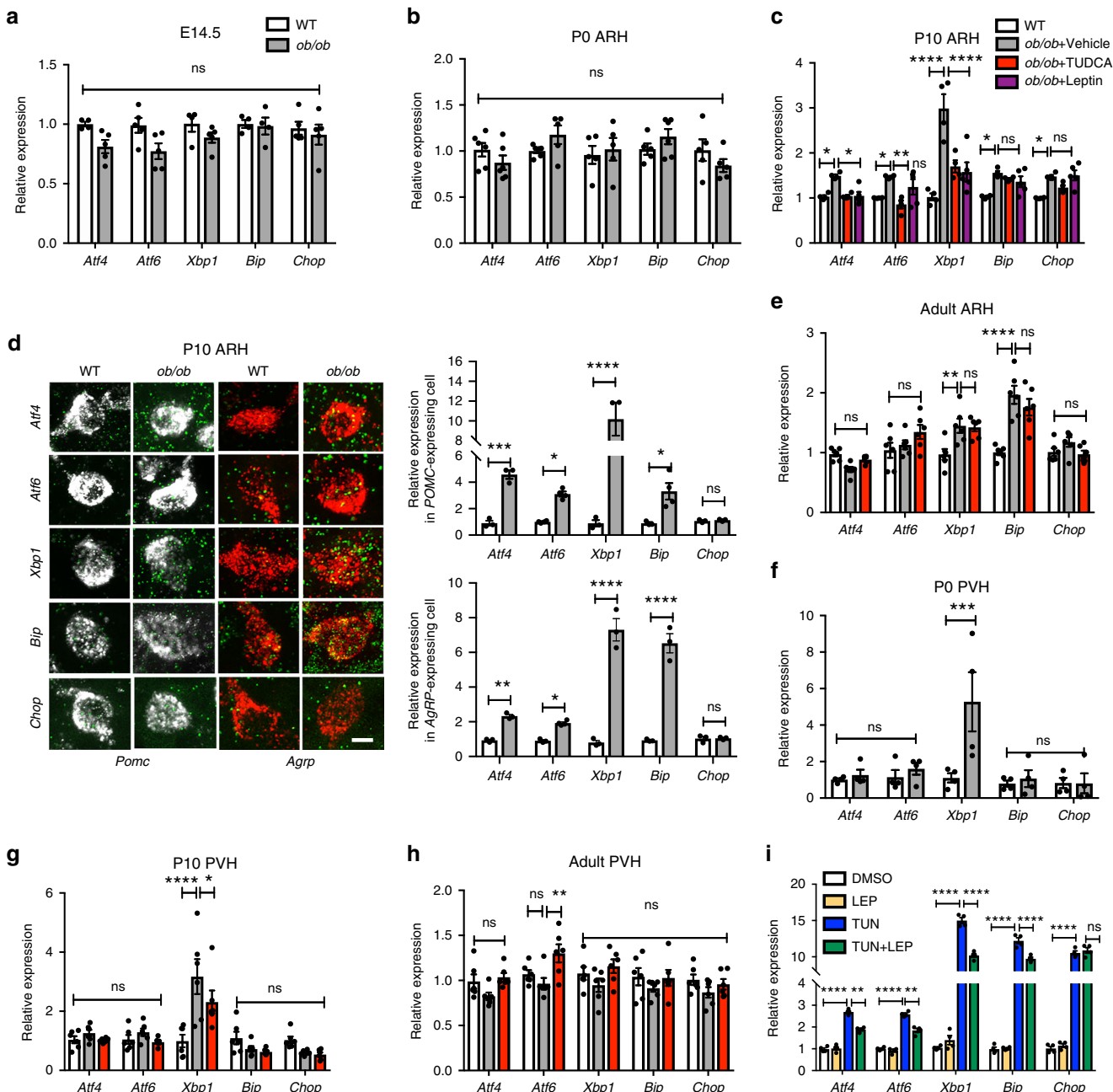

**Fig. 1 Leptin deficiency increases endoplasmic reticulum stress markers in the developing hypothalamus.** Relative expression of activating transcription factor 4 (*Atf4*), 6 (*Atf6*), X-box binding protein (*Xbp1*), glucose regulated protein GRP78 (referred to as *Bip*), and CCAAT-enhancer-binding protein homologous protein (*Chop*) mRNA **a** in the hypothalamus of embryonic day (E)14.5 mice (*n* = 4–5 per group) and **b** in the arcuate nucleus (ARH) of postnatal day (P) 0 wild-type (WT) and leptin-deficient (*ob/ob*) mice (*n* = 5–6 per group). **c** Relative expression of *Atf4, Atf6, Xbp1, Bip,* and *Chop* mRNA in the ARH of P10 WT mice and *ob/ob* mice treated neonatally either vehicle or tauroursodeoxycholic acid (TUDCA) or leptin (WT, *ob/ob* + Vehicle, *ob/ob* + TUDCA *n* = 4, *ob/ob* + Leptin: *n* = 5 per group). **d** Representative images and quantification of *Atf4, Atf6, Xbp1, Bip,* and *Chop* mRNA (green) in arcuate pro-opiomelanocortin (*Pomc*)- (white) and agouti-related peptide (*Agrp*) (red) mRNA-expressing cells of WT and *ob/ob* mice at P10 (*n* = 3–4 per group). **e** Relative expression of *Atf4, Atf6, Xbp1, Bip,* and *Chop* mRNA in the ARH of 10-week-old adult WT mice and *ob/ob* mice treated neonatally either vehicle or TUDCA (*n* = 6 per group). Relative expression of *Atf4, Atf6, Xbp1, Bip,* and *Chop* mRNA in the paraventricular nucleus (PVH) of **f** P0 (*n* = 4 per group), **g** P10 (*n* = 6 per group), and **h** 10-week-old adult WT mice and *ob/ob* mice treated neonatally either vehicle or TUDCA (*n* = 6 per group). **i** Relative expression of *Atf4, Atf6, Xbp1, Bip,* and *Chop* mRNA in hypothalamic mHypoE-N43/5 cell lysates treated with dimethyl sulfoxide (DMSO, control) or leptin (LEP, 100 ng/ml) or tunicamycin (TUN, 0.1 µg/ml) or TUN + LEP for 5 h (*n* = 4 per group). Error bars represent the SEM. *$P \leq 0.05$, **$P \leq 0.01$, ***$P \leq 0.001$, and ****$P \leq 0.0001$ versus *ob/ob* mice (**d**, **f**), *$P \leq 0.05$, **$P \leq 0.01$, ****$P \leq 0.0001$ versus vehicle-injected *ob/ob* mice (**c**, **e**, **g**, **h**). **$P \leq 0.01$ and ****$P \leq 0.0001$ versus tunicamycin treated group (**i**). Statistical significance was determined using two-way ANOVA followed by Tukey's Multiple Comparison test (**a–i**). Scale bar, 5 µm (**d**). Source data are provided as a Source Data file.

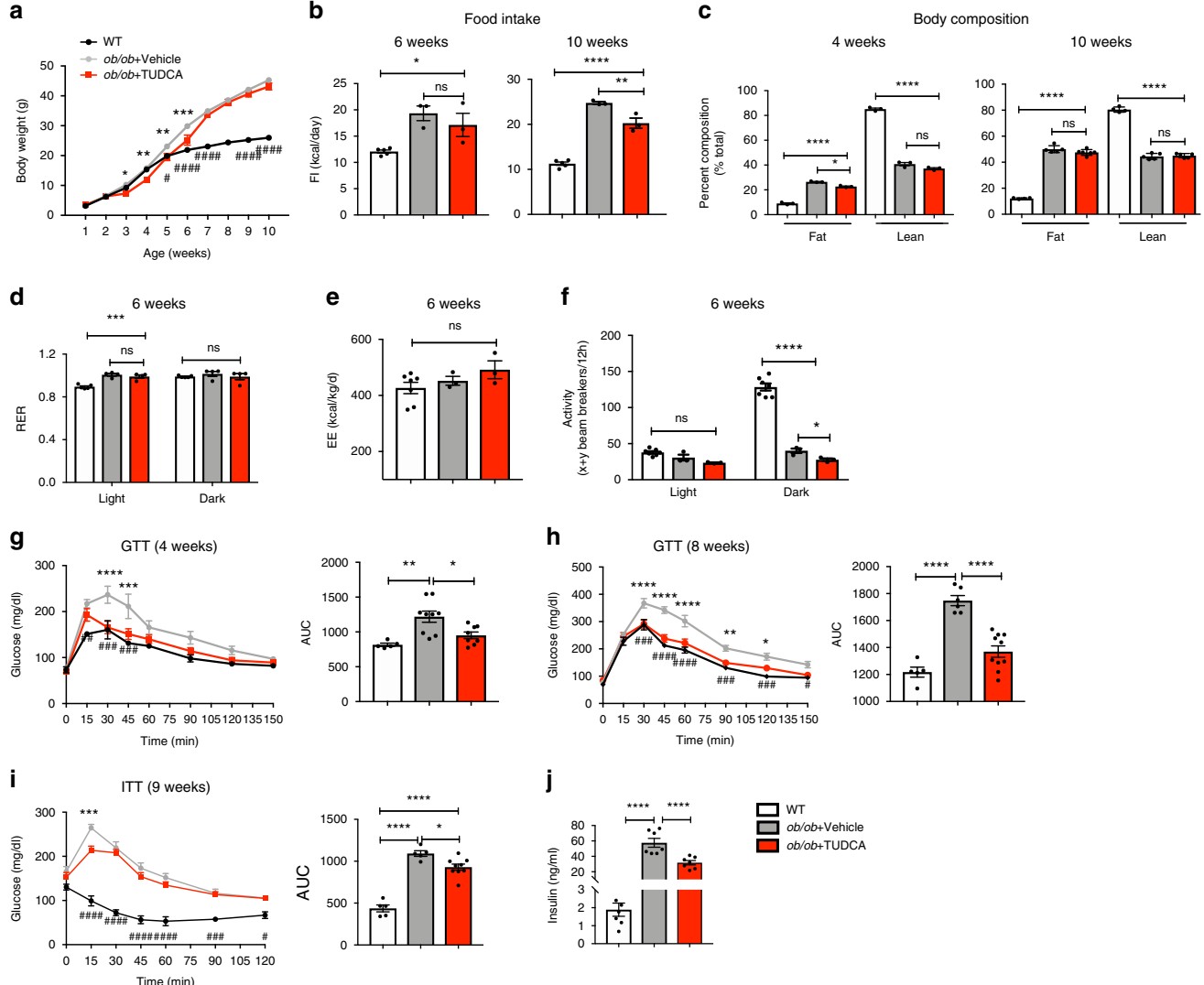

**Fig. 2 Neonatal TUDCA treatment causes long-term beneficial metabolic effects in *ob/ob* mice. a** Growth curves of wild-type (WT) mice and leptin-deficient (*ob/ob*) mice treated neonatally with vehicle or tauroursodeoxycholic acid (TUDCA) ($n = 8$ per group). **b** Food intake (WT: $n = 5$, *ob/ob* + Vehicle, *ob/ob* + TUDCA: $n = 4$ per group) and **c** body composition of 4- ($n = 3$ per group) and 10-week-old ($n = 4$–5 per group) WT mice and *ob/ob* mice treated neonatally with vehicle or TUDCA. **d** Respiratory exchange ratio (WT: $n = 5$, *ob/ob* + Vehicle, *ob/ob* + TUDCA: $n = 4$ per group), **e** energy expenditure (WT: $n = 7$, *ob/ob* + Vehicle, *ob/ob* + TUDCA: $n = 3$ per group), and **f** locomotor activity of 6-week-old WT mice and *ob/ob* mice treated neonatally with vehicle or TUDCA (WT: $n = 7$, *ob/ob* + Vehicle, *ob/ob* + TUDCA: $n = 3$ per group). **g, h** Glucose tolerance tests (GTT) and area under the curve (AUC) quantification of 4-week-old (WT: $n = 5$, *ob/ob* + Vehicle, *ob/ob* + TUDCA: $n = 8$ per group) (**g**) and 8-week-old (WT: $n = 5$, *ob/ob* + Vehicle: $n = 6$, *ob/ob* + TUDCA: $n = 10$ per group) (**h**) WT mice and *ob/ob* mice treated neonatally with vehicle or TUDCA ($n = 5$–10 per group). **i** Insulin tolerance test (ITT) and area under the ITT curve (AUC) of 9-week-old WT mice and *ob/ob* mice treated neonatally with vehicle or TUDCA (WT, *ob/ob* + Vehicle: $n = 5$, *ob/ob* + TUDCA: $n = 9$ per group). **j** Serum insulin levels of 10- to 12-week-old fed WT mice and *ob/ob* mice treated neonatally with vehicle or TUDCA ($n = 7$ per group). Error bars represent the SEM. *$P < 0.05$, **$P \leq 0.01$, ***$P \leq 0.001$, and ****$P \leq 0.0001$ versus vehicle-injected *ob/ob* mice (**a, g–i**), #$P < 0.05$, ##$P \leq 0.01$, ###$P \leq 0.001$, and ####$P \leq 0.0001$ *versus* WT mice (**a, g–i**). *$P < 0.05$, ***$P \leq 0.001$, and ****$P \leq 0.0001$ versus WT mice and *$P < 0.05$, **$P \leq 0.01$, and ****$P \leq 0.0001$ versus vehicle-injected *ob/ob* mice (**b–f, j**). Statistical significance between groups was determined using two-tailed Student's *t* test (**f**), one-way ANOVA (**b, c, e, f, j**) and AUCs in **g–j** and two-way ANOVA (**a, d, g–i**) followed by Tukey's multiple comparison test. Source data are provided as a Source Data file.

light phase appeared comparable between control and TUDCA-treated *ob/ob* mice (Fig. 2d–f). However, locomotor activity during the dark phase was slightly reduced in TUDCA-treated mice (Fig. 2f). To examine whether relieving ER stress neonatally also had consequences on glucose homeostasis, we performed glucose- and insulin-tolerance tests. When exposed to a glucose or an insulin challenge, adult *ob/ob* mice neonatally treated with TUDCA displayed improved glucose and insulin tolerance, respectively, compared to control *ob/ob* mice (Fig. 2g–i). In

addition, neonatal TUDCA treatment reduced insulin levels in adult fed *ob/ob* mice (Fig. 2j).

Together, these data indicate that relieving ER stress neonatally in *ob/ob* mice has long-term metabolic consequences.

**Neonatal TUDCA treatment corrects POMC axonal projections.** During development, axonal projections ascend from the ARH to reach their target nuclei, including the PVH, during the 2nd week of postnatal life[14]. We previously reported that leptin

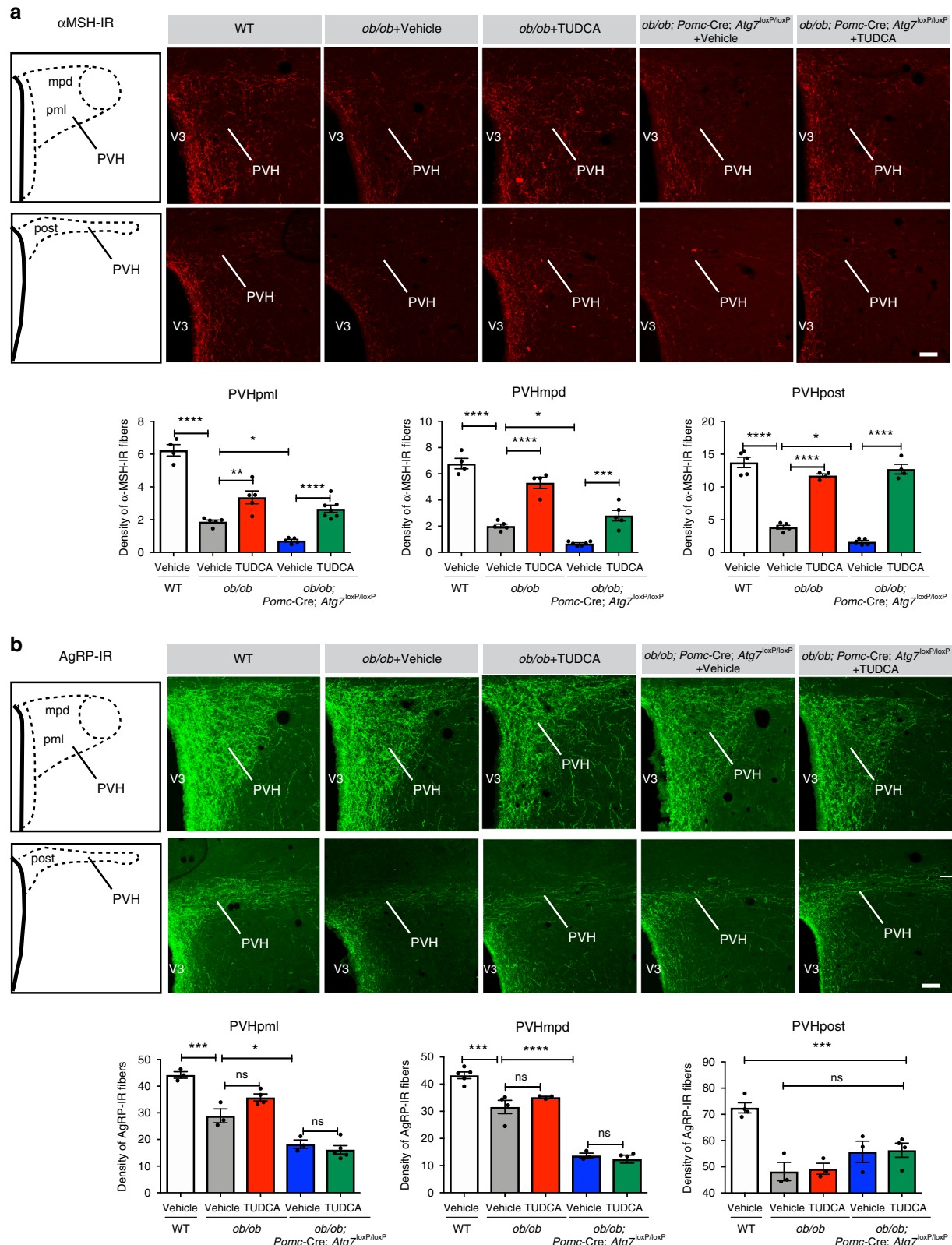

exerts a neurotrophic effect on arcuate circuits and that the development of POMC and AgRP neuronal projections are disrupted in *ob/ob* mice[7]. To determine whether neonatal TUDCA treatment influences the organization of ARH circuits, we performed immunostaining for αMSH and AgRP in animals treated with TUDCA or vehicle between P4 and P16. As previously reported[7,15], vehicle-treated *ob/ob* mice display a marked

reduction in the density of both αMSH- and AgRP-immunoreactive fibers innervating the neuroendocrine and pre-autonomic parts of the PVH (Fig. 3a, b). Neonatal TUDCA treatment in *ob/ob* mice increases the density of αMSH-immunoreactive projections innervating the dorsal component of the medial parvicellular of the PVH, the lateral magnocellular part of the PVH, and the posterior part of the PVH (Fig. 3a).

**Fig. 3 Neonatal tauroursodeoxycholic acid treatment improves pro-opiomelanocortin axonal projections. a, b** Representative images and quantification of the density of **a** α-melanocyte-stimulating hormone (αMSH)—(red) and **b** agouti-related peptide (AgRP)—(green) immunoreactive fibers innervating the neuroendocrine (PVHpml and PVHmpd) and preautonomic (postPVH) compartments of the PVH of 10- to 12-week-old wild-type (WT) mice, leptin-deficient (*ob/ob*) mice treated neonatally with vehicle or tauroursodeoxycholic acid (TUDCA), and *ob/ob; Pomc*-Cre; *Atg7*loxP/loxP mice treated neonatally with vehicle or TUDCA ($n = 3$–6 per group). Error bars represent the ±SEM. *$P \leq 0.05$, **$P < 0.01$, ***$P \leq 0.001$, and ****$P \leq 0.0001$ versus all groups. Statistical significance was determined by one-way ANOVA followed by Tukey's multiple comparison test (**a**, **b**). PVH, paraventricular nucleus of the hypothalamus; PVH, paraventricular nucleus; PVHmpd, dorsal component of the medial parvicellular PVH; PVHpml, lateral magnocellular part of the PVH; post PVH, posterior part of the PVH; V3, third ventricle. Scale bar, 50 μm (**a**, **b**). Source data are provided as a Source Data file.

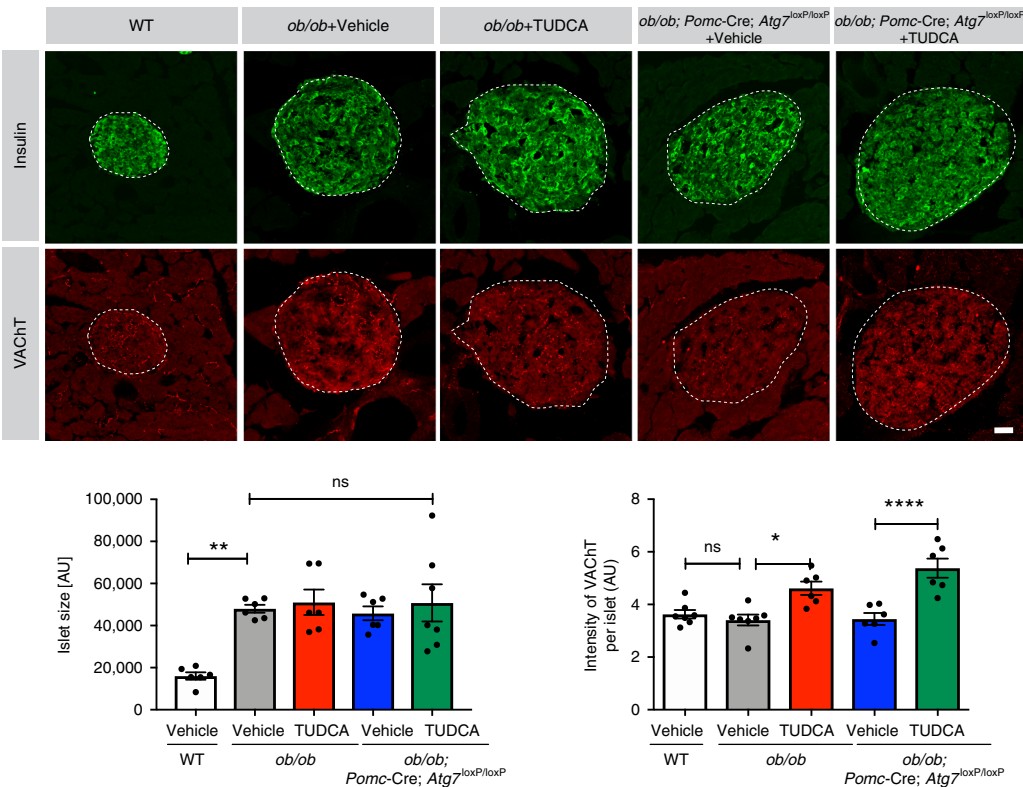

**Fig. 4 Neonatal tauroursodeoxycholic acid treatment improves parasympathetic innervation in the pancreas.** Representative images and quantification of islet size and the density of vesicular acetylcholine transporter(VAChT)-immunoreactive fibers (red) in the insulin+ islets (green) of 10- to 12-week-old wild-type (WT) mice, leptin-deficient (*ob/ob*) mice treated neonatally with vehicle or tauroursodeoxycholic acid (TUDCA), and *ob/ob; Pomc*-Cre; *Atg7*loxP/loxP mice treated neonatally with vehicle or TUDCA ($n = 6$–7 per group). Error bars represent the SEM. *$P \leq 0.05$, **$P < 0.01$, and ****$P \leq 0.0001$ versus all groups. Statistical significance was determined by one-way ANOVA followed by Tukey's multiple comparison test. Scale bar, 100 μm. Source data are provided as a Source Data file.

However, neonatal TUDCA treatment had no effects on AgRP-immunopositive projections (Fig. 3b).

The posterior PVH sends descending projections to regions of the brainstem and spinal cord that regulate the activity of the autonomic nervous system, which plays an important role in glucose homeostasis[15–17]. In addition, during development, leptin appears to play a critical role in the parasympathetic innervation of pancreatic β cells[18]. Given that TUDCA treatment in *ob/ob* neonates improves POMC projections to the preautonomic part of the PVH and has beneficial effects on glucose homeostasis, we also analyzed the parasympathetic innervation of pancreatic islets by performing immunostaining for vesicular acetylcholine transporter (VAChT)[18,19]. The density of VAChT-immunopositive fibers innervating pancreatic β cells was significantly increased in TUDCA-treated *ob/ob* mice compared to those in control animals (Fig. 4). Islet size was increased in *ob/ob* mice compared to that in WT mice, but neonatal TUDCA treatment did not affect islet size (Fig. 4).

Collectively, these data indicate that enhancing ER capacity by neonatal TUDCA treatment improves αMSH neuronal projections and has long-term beneficial metabolic outcomes.

**ER stress and leptin deficiency trigger autophagy.** The ER is not only involved in protein synthesis and maturation but also in the formation of autophagosomes[20]. Moreover, cells exposed to ER stress undergo the UPR to avoid apoptosis, but also activate autophagy[21]. We therefore evaluated whether ER stress induces autophagy in hypothalamic cells by exposing mHypoE-N43/5 cells to the ER stress inducer tunicamycin and measured autophagy-related gene expression. Tunicamycin treatment in vitro increased the level of the autophagy-related genes *Atg5*, *Atg7*, *Atg12*, and *p62* (Fig. 5a). Furthermore, mHypoE-N43/5 cells exposed to tunicamycin displayed elevated levels of the microtubule-associated protein light chain 3 (LC3B), a common marker of autophagosomes[22] (Fig. 5b).

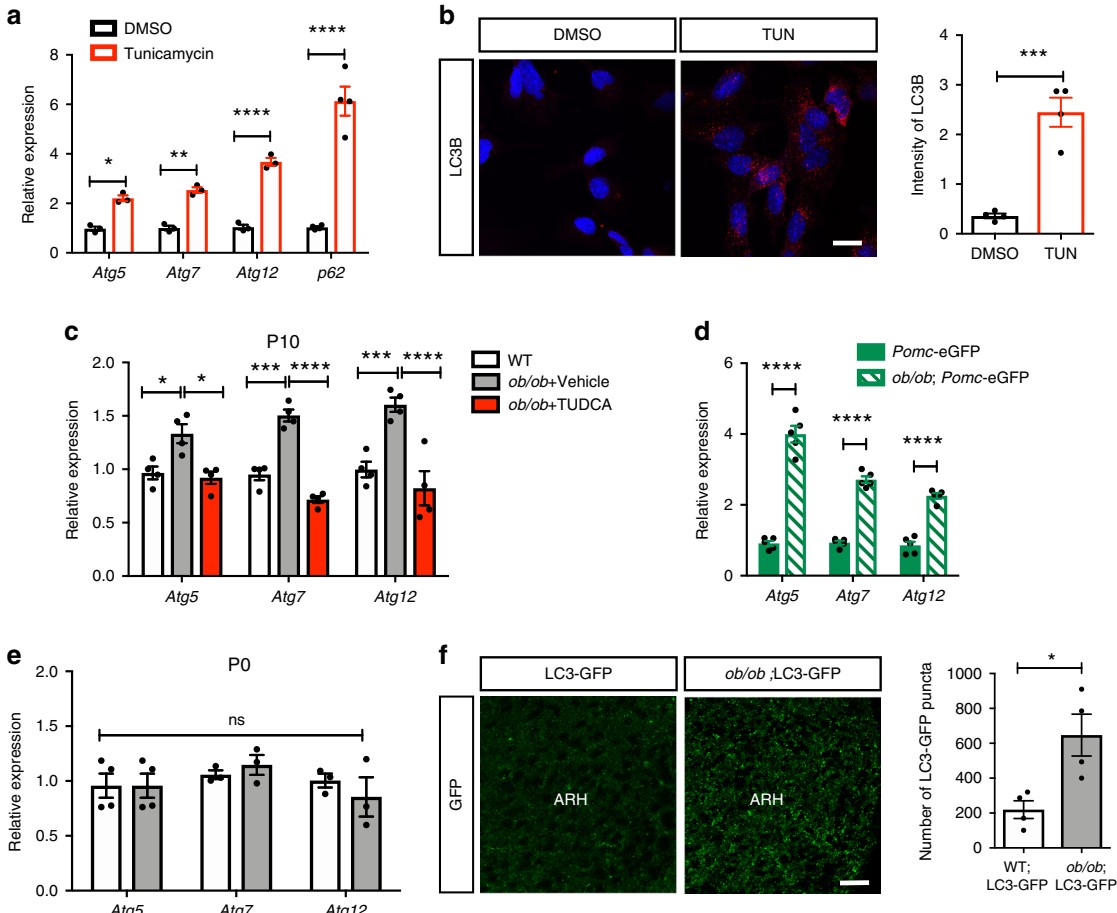

**Fig. 5 Induction of ER stress and leptin deficiency promote autophagy genes. a** Relative expression of autophagy-related protein (*Atg*) 5, *Atg7*, and *Atg12* mRNA in mouse hypothalamic mHypoE-N43/5 treated with DMSO (control) or tunicamycin (0.1 μg/ml) for 5 h ($n = 3$–4 per group). **b** Representative images and quantification of microtubule-associated protein light chain 3 (LC3B)-immunoreactive puncta (red) in mouse hypothalamic mHypoE-N43/5 cells treated with DMSO (control) or tunicamycin (0.1 μg/ml) for 5 h ($n = 4$ per group). **c** Relative expression of *Atg5*, *Atg7*, and *Atg12* mRNA in the arcuate nucleus (ARH) of postnatal day (P)10 wild-type (WT) mice and leptin-deficient (*ob/ob*) mice treated with either vehicle or tauroursodeoxycholic acid (TUDCA) neonatally ($n = 4$ per group). **d, e** Relative expression of *Atg5*, *Atg7*, and *Atg12* mRNA in (**d**) sorted *Pomc*-GFP cells ($n = 5$ per group) and **e** the arcuate nucleus (ARH) of P0 WT (white bars) and *ob/ob* (gray bars) mice ($n = 3$–4 per group). **f** Representative images and quantification of LC3-GFP⁺ puncta (green) in the ARH of P10 WT (white bars) and *ob/ob* (gray bars) mice ($n = 4$ per group). Error bars represent the SEM. *$P \leq 0.05$, **$P \leq 0.01$, ***$P \leq 0.001$, and ****$P \leq 0.0001$ *versus* DMSO-treated cells (**a, b**); *$P \leq 0.05$, ***$P \leq 0.001$, and ****$P \leq 0.0001$ versus vehicle-injected *ob/ob* mice (**c**); ****$P \leq 0.0001$ versus Pomc-GFP mice (**d**); *$P \leq 0.05$ versus LC3-GFP mice (**f**). Statistical significance between groups was determined using two-tailed Student's *t* test (**b, f**) and two-way ANOVA (**a, c, d, e**) followed by Tukey's multiple comparison test. ARH, arcuate nucleus of the hypothalamus. Scale bars, 100 μm (**b**), and 20 μm (**f**). Source data are provided as a Source Data file.

In vivo, *Atg5*, *Atg7*, and *Atg12* mRNAs were upregulated in the arcuate nucleus of leptin-deficient neonates at P10, and this effect was reversed by neonatal TUDCA treatment (Fig. 5c). Increases in *Atg5*, *Atg7*, and *Atg12* mRNA expression was specifically observed in POMC neurons of P10 *ob/ob* neonates (Fig. 5d). However, the induction of autophagy-related genes was not observed in newborn *ob/ob* mice (Fig. 5e). We also crossed *ob/ob* mice with transgenic mice in which LC3 was fused to the green fluorescent protein (GFP)[23] and found that leptin deficiency increased the density of LC3-GFP puncta in the ARH at P10 (Fig. 5f).

**Metabolic defects in *ob/ob*; *Pomc*-Cre; *Atg7*[loxP/loxP] mice.** To further investigate the relationship between leptin deficiency, ER stress and autophagy in vivo, we generated *ob/ob* mice that lack the autophagy gene *Atg7* in POMC neurons (*ob/ob*; *Pomc*-Cre; *Atg7*[loxP/loxP] mice). The rationale for focusing on POMC neurons, specifically, is because we found that TUDCA treatment

only rescues POMC projections in *ob/ob* mice (see results above), and we previously reported that *Atg7* in POMC neurons is required for normal metabolic regulation and neuronal development[24]. We confirmed that these mice are indeed autophagy-defective by analyzing the expression of the polyubiquitin-binding protein p62, which links ubiquitinated proteins to the autophagy apparatus[25]. We found that *ob/ob*; *Pomc*-Cre; *Atg7*[loxP/loxP] mice displayed higher levels of p62-IR in the ARH compared to WT and *ob/ob* mice (Fig. 6a). The *ob/ob*; *Pomc*-Cre; *Atg7*[loxP/loxP] mice were born normal and survived to adulthood. In addition, these mutant mice had body weights indistinguishable from those of their *ob/ob* control littermates during the preweaning period and in adulthood (Fig. 6b, c). However, when exposed to a glucose challenge, *ob/ob*; *Pomc*-Cre; *Atg7*[loxP/loxP] mice displayed impaired glucose tolerance compared to *ob/ob* mice (Fig. 6d). Serum insulin levels appeared normal in fed *ob/ob*; *Pomc*-Cre; *Atg7*[loxP/loxP] mice (Fig. 6e). In addition, food intake and fat mass were increased in 10-week-old and

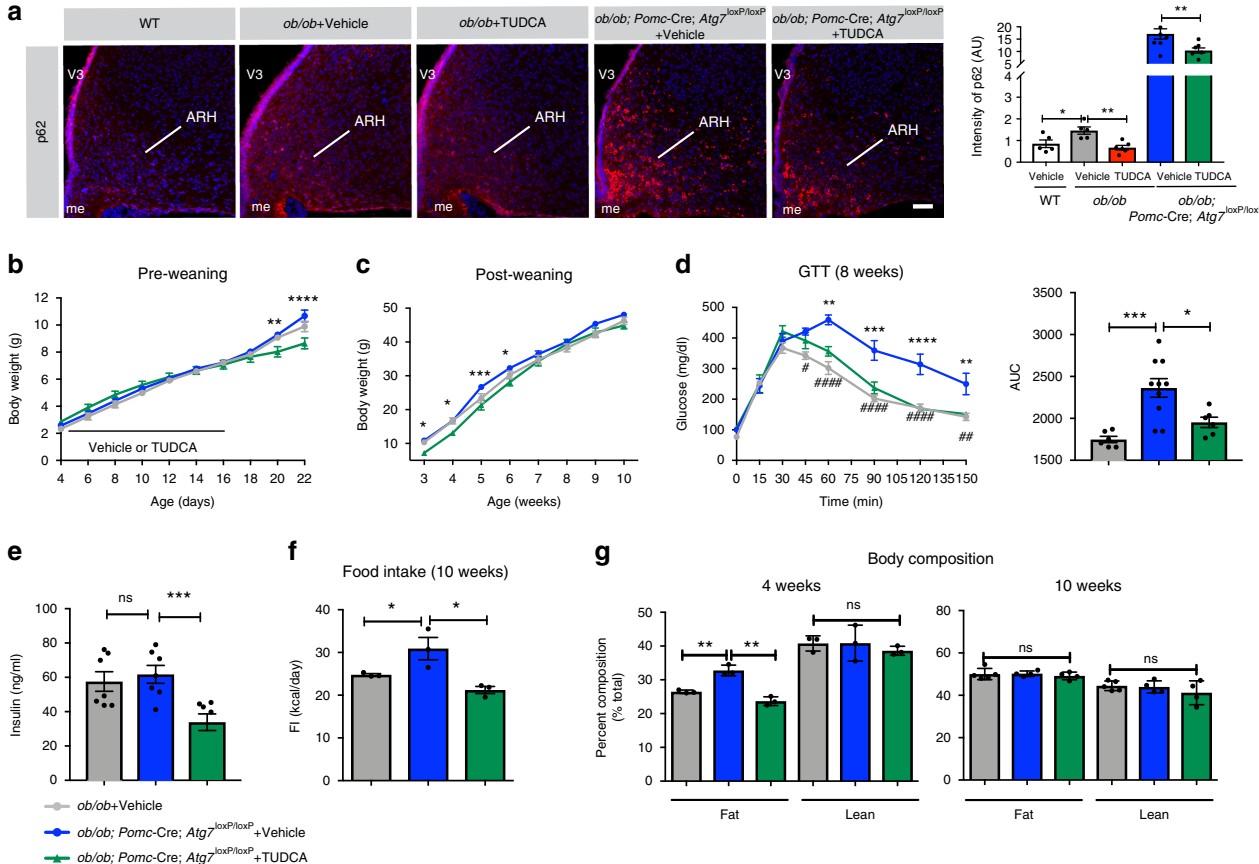

**Fig. 6 Neonatal tauroursodeoxycholic acid treatment ameliorates metabolic defects in *ob/ob; Pomc*-Cre; *Atg7*^loxP/loxP^ mice. a** Representative images and quantification of ubiquitin-binding protein (p62) immunoreactivity (red) in the arcuate nucleus (ARH) of 10-week-old wild-type (WT) mice, leptin-deficient (*ob/ob*) mice treated neonatally with vehicle or tauroursodeoxycholic acid (TUDCA), and *ob/ob; Pomc*-Cre; *Atg7*^loxP/loxP^ mice treated neonatally with vehicle or TUDCA (n = 5–7 per group). **b** Pre- (ob/ob, *ob/ob; Pomc*-Cre; *Atg7*^loxP/loxP^ + Vehicle: n = 9, *ob/ob; Pomc*-Cre; *Atg7*^loxP/loxP^ + TUDCA: n = 8 per group) and (**c**) post-weaning growth curves of *ob/ob* mice and *ob/ob; Pomc*-Cre; *Atg7*^loxP/loxP^ mice treated neonatally with vehicle or TUDCA (ob/ob, *ob/ob; Pomc*-Cre; *Atg7*^loxP/loxP^ + Vehicle: n = 7, *ob/ob; Pomc*-Cre; *Atg7*^loxP/loxP^ + TUDCA: n = 6 per group). **d** Blood glucose concentration after intraperitoneal injection of glucose and area under the glucose tolerance test (GTT) curve (AUC) (ob/ob, *ob/ob; Pomc*-Cre; *Atg7*^loxP/loxP^ + TUDCA: n = 6, *ob/ob; Pomc*-Cre; *Atg7*^loxP/loxP^ + Vehicle: n = 10 per group) and **e** serum insulin levels of 8-week-old fed *ob/ob* mice and *ob/ob; Pomc*-Cre; *Atg7*^loxP/loxP^ mice treated neonatally with vehicle or TUDCA (n = 7 per group). **f** Food intake of 10-week-old *ob/ob* mice and *ob/ob; Pomc*-Cre; *Atg7*^loxP/loxP^ mice treated neonatally with vehicle or TUDCA (n = 3–5 per group). **g** Body composition of 4- (n = 3 per group) and 10-week-old *ob/ob* mice and *ob/ob; Pomc*-Cre; *Atg7*^loxP/loxP^ mice treated neonatally with vehicle or TUDCA (n = 4–5 per group). Error bars represent the SEM. *$P \leq 0.05$, and **$P < 0.01$ versus all groups (**a**), *$P \leq 0.05$, **$P < 0.01$, ***$P \leq 0.001$, and ****$P \leq 0.0001$ versus vehicle-treated *ob/ob; Pomc*-Cre; *Atg7*^loxP/loxP^ mice (**b**–**g**), #$P < 0.05$, ##$P \leq 0.01$, ###$P \leq 0.001$, and ####$P \leq 0.0001$ versus WT mice (**d**). Statistical significance between groups was determined using one-way ANOVA (**a**, AUC in **d**, **e**–**g**) and two-way ANOVA (**b**–**d**) followed by Tukey's multiple comparison test. ARH, arcuate nucleus of the hypothalamus; me, median eminence; V3, third ventricle. Scale bar, 50 μm (**a**). Source data are provided as a Source Data file.

4-week-old, *ob/ob; Pomc*-Cre; *Atg7*^loxP/loxP^ mice (Fig. 6f, g), but the body composition became normal in 10-week-old mice (Fig. 6g). We also treated *ob/ob; Pomc*-Cre; *Atg7*^loxP/loxP^ mice with TUDCA daily from P4 to P16 and found that neonatal TUDCA treatment reduced arcuate p62-IR, body weight, serum insulin levels, food intake, and body composition and improved glucose tolerance (Fig. 6a–g).

Structurally, the loss of *Atg7* expression in POMC neurons in *ob/ob* mice further disrupted the density of αMSH and AgRP projections innervating the PVH (Fig. 3a, b) but had no effect on the parasympathetic innervation of pancreatic islets (Fig. 4) or POMC cell numbers (Supplementary Fig. 2). However, neonatal TUDCA treatment in *ob/ob; Pomc*-Cre; *Atg7*^loxP/loxP^ mice increased the density of αMSH projections to the PVH and VAChT projections to pancreatic islets but had no effects on AgRP projections to the PVH (Fig. 3a–b and 4).

Collectively, these data indicate that the loss of autophagy in POMC neurons exacerbates the metabolic and neurodevelopmental

abnormalities observed in *ob/ob* mice and that relieving ER stress neonatally improves these outcomes.

## Discussion

Previous studies have associated defective ER stress with metabolic disorders. For example, adult *ob/ob* mice exhibit elevated levels of ER stress in the liver, adipose tissue and hypothalamus, and pharmacological inhibition of ER stress during adulthood restores normal glycemia and insulin sensitivity in *ob/ob* mice[9,11,12]. However, much of the effort has been focused on the induction and role of ER stress during the adult period, but the importance of early life ER stress on metabolic regulation and brain development remains unknown. In the present study, we show that increased ER stress in *ob/ob* mice occurs as early as the early postnatal period and that relieving ER stress neonatally has long-term consequences on body weight, food intake, and glucose regulation, with a more pronounced effect on glucose tolerance

than insulin sensitivity. We also found that the neonatal TUDCA administration improves defective hypothalamic POMC axonal projections in *ob/ob* mice.

A strong link has been reported between ER stress and central leptin sensitivity[9,12]. Chronic treatment of genetically obese *ob/ob* mice or diet-induced obese mice with the chemical chaperone 4-phenyl butyric acid (PBA) enhances leptin-induced weight loss and hypophagia associated with improved hypothalamic leptin receptor signaling[12]. Moreover, the pharmacological induction and genetic loss of the ER stress function in the brain block hypothalamic leptin-induced STAT3 activation[12]. Notably, elevated ER stress is observed as early as 4 days after high-fat feeding, i.e., before the mice become obese, suggesting that factors other than obesity cause ER stress in diet-induced obese mice[26]. Consistent with this idea, we also found that *ob/ob* mice display higher levels of ER stress in the hypothalamus, adipose, and liver tissues at P10, i.e., when these mice are still lean[27]. Leptin appears to be a critical factor involved in ER stress because we found that chronic treatment of *ob/ob* neonates with leptin restores normal levels of hypothalamic ER stress, and the exposure of hypothalamic cells in vitro to leptin reduced tunicamycin-induced ER stress.

Leptin has been involved in a variety of biological processes, including the development of neural circuits involved in metabolic regulation. Leptin-deficient mice have disrupted the development of projections from the arcuate nucleus and hindbrain neurons[7,18]. Remarkably, exogenous administration of leptin in *ob/ob* neonates reverses the abnormal development of arcuate circuits, but this effect is loss in adult mice, suggesting the importance of the early postnatal period for the neutrophic action of leptin[7,8,15]. The cellular mechanisms underlying the effects of leptin on postnatal hypothalamic development remain largely elusive. The present study provides evidence that neonatal ER stress could contribute to the metabolic and neurodevelopmental perturbations observed in a context of leptin deficiency. The administration of the ER stress-relieving bile acid TUDCA during a critical period of postnatal development causes long-term beneficial effects on body weight, food intake, adiposity, and glucose homeostasis and improves the development of POMC projections to both the neuroendocrine and parasympathetic compartments of the PVH in *ob/ob* mice. Intriguingly, neonatal TUDCA treatment does not influence AgRP circuits even though we found higher ER stress levels in arcuate AgRP neurons in P10 *ob/ob* mice. These data suggest that neonatal TUDCA treatment acts more effectively on POMC neurons than AgRP neurons to influence axon growth and/or that the disrupted development of AgRP projections in independent on ER stress. Previous studies have also linked ER stress with abnormal POMC maturation. For example, pharmacological inducers of ER stress causes a reduction in αMSH secretion, and the treatment of diet-induced obese animals with TUDCA reverses defects in αMSH secretion but do not influence NPY peptide[28]. The mechanisms underlying the effects of ER stress on POMC maturation appear to involve abnormal post-tposttranslational processing of POMC by decreasing the expression of prohormone convertase 2, an enzyme responsible for processing POMC-derived peptides[28].

In the present study, we found high levels of Xbp1 in the arcuate nucleus of *ob/ob* neonates that is associated with lifelong metabolic perturbations. This data contrast with previous studies revealing that genetic induction of Xbp1s in POMC neurons protects against diet-induced obesity and improves leptin and insulin sensitivity[10]. However, these genetic manipulations are restricted to one cell type and these effects could be canceled out or oppositely regulated with the recruitment of more cell populations, such as when animals are treated with TUDCA. In addition, Williams and colleagues[10] examined selective overexpression of a single UPR factor (i.e., Xbp1). However, in disease states (e.g., in *ob/ob* mice) these factors are not commonly activated alone and multiple ER stress pathways are upregulated. For example, we found that *Atf4*, *Atf6*, *Xbp1*, *Bip*, and *Chop* are upregulated in the arcuate nucleus of *ob/ob* neonates. Moreover, Ozcan et al.[12] reported that mice genetically deficient in *Xbp1* in the CNS displayed higher levels of other ER stress pathways suggesting that compensatory mechanisms may occur in mice with deletion of a single ER stress pathway.

The ER stress pathway can also regulate multiple steps of the autophagy process, such as autophagy induction, vesicle nucleation, and elongation of the phagophore[20]. Hypothalamic autophagy has recently been implicated in metabolic regulation. For example, deletion of the autophagy-related gene *Atg7* in the mediobasal hypothalamus causes metabolic perturbations[29]. In addition, mice lacking *Atg7* specifically in POMC neurons display higher body weights, hyperphagia, leptin resistance, and impaired glucose tolerance associated with disrupted development of POMC axonal projections[24,30,31]. Similarly, the loss of *Atg12* in POMC neurons promotes weight gain and perturbations in glucose homeostasis induced by high-fat feeding[32]. In contrast, deficiency in *Atg4* in POMC neurons or in *Atg7* in AgRP neurons causes leanness, suggesting that autophagy induces distinct physiological effects based on the pathways and cell types[33,34]. However, a link between leptin, ER stress and autophagy has not been reported. Here, we showed that *ob/ob* neonates display elevated levels of autophagy-related genes and associated p62 accumulation in the arcuate nucleus, which can be normalized upon TUDCA treatment. Moreover, genetic deletion of *Atg7* in POMC neurons of *ob/ob* mice worsens their glucose homeostasis, adiposity, hyperphagia, and POMC neuronal projections, and neonatal TUDCA treatment causes long-term improvements in this phenotype. These data suggest that ER stress induced by leptin deficiency prompts the activation of autophagy during development leading to long-term effects on metabolism and hypothalamic development.

## Methods

**Animals**. All animal procedures were conducted in compliance with all relevant ethical regulations for animal testing and research. In addition, the experiments described in the present study were approved by the Institutional Animal Care and Use Committee of the Saban Research Institute of the Children's Hospital of Los Angeles. The animals were housed under specific pathogen-free conditions, maintained in a temperature-controlled room with a 12 h light/dark cycle, and provided ad libitum access to water and standard laboratory chow (Special Diet Services). To generate leptin and POMC-specific Atg7 knockout (*ob/ob*; *Pomc*-Cre; *Atg7*^loxP/loxP^) mice, *Pomc*-Cre mice (Jax mice stock #5965)[35] were mated to mice carrying a loxP-flanked *Atg7* allele (*Atg7*^loxP/loxP^), which was kindly provided by Pr. Komatsu[36]. *Pomc*-Cre; *Atg7*^loxP/loxP^ mice were then crossed with *ob/−* mice (Jax mice stock #632). The breeding colonies were maintained by mating *ob/−*; *Pomc*-Cre; *Atg7*^loxP/+^ mice to *ob/−*; *Atg7*^loxP/loxP^ mice. LC3-GFP mice were kindly provided by Pr. Mizushima[23] and were also crossed with *ob/−* mice to generate LC3-GFP; *ob/ob* mice. All mice were generated in a C57BL/6 background, and only male mice were studied. Mouse genotyping primers are shown in Supplementary Table 1.

**Neonatal TUDCA treatment**. The *ob/ob* mice were injected daily with TUDCA (Millipore, 150 mg/kg/day) from P4 to P16. Controls received injections with an equivalent volume of vehicle (0.9% NaCl).

**Tissue collection**. The hypothalamus of E14.5 mouse embryos as well as the ARH and PVH of P0, P10 and 10-week-old mice (*n* = 6 per group) were dissected under a stereomicroscope. Liver and adipose tissues were dissected from P10 or 10-week-old mice.

**Cell sorting**. The hypothalami of P10 *Pomc*-eGFP and *ob/ob*; *Pomc*-eGFP mice (*n* = 5 per group) were microdissected under a stereomicroscope and enzymatically dissociated using the Papain Dissociation System (Worthington) according to the manufacturer's instructions. Fluorescence-activated cell sorting (FACS) was performed using a BD FACS Aria II Cell Sorter to sort GFP⁺ cells (Supplementary

Fig. 3). Non-fluorescent cells obtained from wild-type mice were used to set the threshold for fluorescence.

**Cell culture and tunicamycin treatment**. The embryonic mouse hypothalamus cell line mHypoE-N43/5 (Cedarlane, cat #CLU127) was cultured in Dulbecco's modified Eagle's medium (Sigma, D5796) supplemented with 10% fetal bovine serum, 100 U/ml penicillin and 100 μg/ml streptomycin at 37 °C in 5% $CO_2$ and a humidified atmosphere. The cells were treated with 0.1 μg/ml tunicamycin (Sigma, T7765) or DMSO (control) in the cell culture medium.

**RT-qPCR analyses**. Total RNA was isolated using the Arcturus PicoPure RNA Isolation Kit (for hypothalamic samples) (Life Technologies) or the RNeasy Lipid Tissue Kit (for peripheral samples) (Qiagen). cDNA was generated with the High-Capacity cDNA Reverse Transcription Kit (Life Technologies). Quantitative real-time PCR was performed using TaqMan Fast Universal PCR Master Mix and the commercially available TaqMan gene expression primers: *Atf4* (Mm00515324_m1), *Atf6* (Mm01295317_m1), *Xbp1* (Mm00457357_m1) *Bip* (Mm00517691_m1), *Chop* (Mm00492097_m1), *Atg5* (Mm01187303_m1), *Atg7* (Mm00512209_m1), *Atg12* (Mm00503201_m1) and *Gapdh* (Mm99999915_g1). mRNA expression was calculated using the $2^{-\Delta\Delta Ct}$ method after normalization to the expression of the *Gapdh* housekeeping gene. All assays were performed using an Applied Biosystems 7900 HT real-time PCR system.

**Physiological measures**. Mice ($n \geq 9$ per group) were weighed every 2 days from P4 to P22 (weaning) and weekly from 4 to 10 weeks of age using an analytical balance. Body composition analysis (fat/lean mass) was performed in 6- and 10-week-old mice ($n = 6$–8 per group) using NMR (Echo MRI). Food intake, $O_2$ and $CO_2$ production, energy expenditure, and locomotor activity (XY) were monitored at 6 weeks of age using a combined indirect calorimetry system (TSE Systems) ($n = 4$–6 per group). The mice were acclimated in monitoring chambers for 2 days, and the data were collected for 3 days. These physiological measures were performed at the Rodent Metabolic Core of Children's Hospital of Los Angeles.

Glucose and insulin tolerance tests (GTT and ITT) were conducted in 4- and 8-week-old mice ($n = 9$–11 per group) through the i.p. injection of glucose (1.5 mg/g body weight) or insulin (2 U/kg body weight) after overnight fasting. Blood glucose levels were measured at 0, 15, 30, 45, 60, 90, 120, and 150 min post-injection.

Serum insulin levels were assayed at 10–12 weeks of age ($n = 7$–8 per group) using an insulin ELISA kit (Millipore).

**Immunohistochemistry**. In all, 10- to 12-week-old mice ($n = 7$–8 per group) were perfused transcardially with 4% paraformaldehyde. The brains were then frozen, sectioned at 30-μm thick, and processed for immunofluorescence using standard procedures[7]. The primary antibodies used for IHC were as follows: sheep anti-αMSH antibody (1:40,000, Millipore, cat #AB5087), rabbit anti-AgRP (1:1,000, Phoenix Pharmaceuticals, cat #H003-53), rabbit anti-LC3B (1:500, Cell Signaling, cat #2775), rabbit anti-p62 (1:1,000, Abcam, cat #ab64134), rabbit anti-GFP (1:1,000, Invitrogen, cat #A-6455), guinea pig anti-insulin (1:500, Abcam, cat #ab7842), and rabbit anti-vesicular acetylcholine transporter (VAChT, 1:500, Synaptic Systems, cat #139103). The primary antibodies were visualized with Alexa Fluor 568 donkey anti-sheep IgG (1:200, Invitrogen, cat #A21099), Alexa Fluor 488 donkey anti-rabbit IgG (1:200, Invitrogen, cat #A21206), Alexa Fluor 488 donkey anti-mouse IgG (1:200, Invitrogen, cat #A21202), Alexa Fluor 568 donkey anti-rabbit IgG (1:200, Invitrogen, cat #A10042), or Alexa Fluor 488 goat anti-guinea pig IgG (1:200, Invitrogen, cat #A11073). The sections were counterstained using bis-benzamide (1:10,000, Invitrogen, cat #H3569) to visualize cell nuclei.

**Multiplex fluorescent in situ hybridization**. P10 mice were perfused with 4% paraformaldehyde. The brains were then frozen, sectioned at 20-μm thick, and processed for multiplex fluorescent in situ hybridization. Commercially available RNAscope Multiplex Fluorescent reagent kits and RNAscope probes (*Agrp* #400711, *Pomc* #314081, *Xbp1* #408911, *Atf4* #405101, *Atf6* #555271, *Bip* #438831, *Chop* #317661) were used for transcript detection (Advanced Cell Diagnostics, Hayward, CA).

**Quantitative analysis of cell numbers and fiber density**. The images were acquired using a Zeiss LSM 710 confocal system equipped with a 20X objective through the ARH (for cell numbers, and p62 and LC3-GFP fluorescence intensity), through the PVHmpd and PVHpml (neuroendocrine) and postPVH (autonomic), and through the pancreas. The average number of cells and density of fibers were analyzed in 2–4 sections. The image analysis was performed using ImageJ analysis software (NIH).

For the quantitative analysis of cell number, POMC$^+$ cells were manually counted. Only cells with corresponding bis-benzamide-stained nuclei were included in our counts.

For the quantitative analysis of fiber density (for αMSH, AgRP, and VAChT fibers in insulin$^+$ pancreatic islets) and the quantification of p62 immunoreactivity, each image plane was binarized to isolate labeled fibers from the background and to compensate for differences in fluorescence intensity. The integrated intensity, which reflects the total number of pixels in the binarized image, was then calculated for each image. This procedure was conducted for each image plane in the stack, and the values for all of the image planes in a stack were summed.

The NIH ImageJ macro, called GFP-LC3 (http://imagejdocu.tudor.lu/author/rkd8/)[37], was used to quantify the number of LC3-GFP puncta.

**Statistical analysis**. All values are represented as the means ± SEM. Statistical analyses were conducted using GraphPad Prism (version 5.0a). Data sets with only two independent groups were analyzed for statistical significance using unpaired two-tailed Student's t test. Data sets with more than two groups were analyzed using one-way analysis of variance (ANOVA) followed by Tukey's multiple comparison test. For statistical analyses of body weight and GTT, we performed two-way ANOVA followed by Tukey's multiple comparison test. Statistically significant outliers were calculated using Grubb's test for outliers. $P \leq 0.05$ was considered statistically significant.

**Reporting summary**. Further information on research design is available in the Nature Research Reporting Summary linked to this article.

## Data availability
Source data for Figs. 1a–i, 2a–j, 3a, b, 4, 5a–f, and 6a–g and Supplementary Figs. 1a–d are provided as a Source Data file. All data is available from the corresponding author upon request.

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

## Acknowledgements
We thank Brad Wanken for his assistance with the analysis of the metabolic studies.

## Author contributions
S.P. conceived, designed, and performed most of the experiments and analyzed the data. A.A. performed and analyzed some of the immunohistochemical experiments. B.C. performed and analyzed some of the physiological experiments. S.G.B. conceived, designed, and supervised the project. S.P. and S.G.B. wrote the paper.

## Competing interests
The authors declare no competing interests.
