## [Peer Review File · Nature Communications]

Reviewers' comments:

Reviewer #1 (Remarks to the Author):

The manuscript by Park and collaborators studies ER stress and autophagy in ob/ob mice during development. They showed increased expression of ER stress- and autophagy-related genes in the hypothalamic arcuate nucleus of neonate ob/ob mice. These effects were specific of POMC neurons and neonatal treatment with the ER stress-relieving drug TUDCA ameliorated the metabolic phenotype of ob/ob mice while selective deletion of Atg7 in POMC neurons worsened the metabolic phenotype of these genetically obese mice.

Overall the study is interesting and the focus of ER stress and autophagy during development is novel.

However, the study described in the manuscript is very hard to follow. The sequence of the data and how they are discussed in relation to the figures is very confusing. Some of the data shown are either not discussed or in discrepancy with other data shown. Pictures should be properly labelled and the overall discussion of the data should be more in-depth.

Specific comments:

Lines 125-126 on page 5 authors stated: "During adulthood leptin deficiency only causes an increase in Atf6 and Xbp1 mRNA expression in the ARH (Fig.1f)". However, Bip also is shown in the figure to be significantly altered while Atf4 seems to be significantly reduced in ob/ob mice.

They are not mentioned and discussed.

In Fig. 1j data on WT and ob/ob mice are a duplication of Fig. 1c.

In Line 143 authors jumped from discussing data shown in Fig. 1k to data shown in Fig.4a. It is unclear why the authors organized the figures in a such confusing manner.

Fig 2a and 2b should be reconciled. It is known that ob/ob mice at birth and until about 6 weeks of age are indistinguishable from the WT control. In agreement with this, in figure 2b the authors show that post-weaning, ob/ob and WT mice have similar body weight from 3 weeks (at weaning) and up to 5 weeks of age. However, in Fig 2a pre-weaning WT mice (20 to 24 days) are significantly lighter than the ob/ob injected with vehicle.

Fig. 2 d show body composition done at 4 weeks of age. However, in the text (line 166) the authors referred to that as done at 6 weeks. Moreover, they stated in the text (line 167-168) that this reduction was not observed in 10 weeks old mice and referred to Fig.2d which, again, show data from 4 weeks old mice.

Fig. 2j reports ITT of 8 weeks old mice. However, in the legend the author reported that mice were 9 weeks old. It is unclear at what age insulin in fig 2k was measured and in what conditions (fed vs fasting).

TUDCA-treated mice showed improved glucose tolerance test (Fig. 2h and i) but they show insulin resistant (Fig. 2j). No comments or discussion of these data is attempted by the authors.

Fig. 3c the outlined of the islet is off in a panel and in general the pictures do not seem to be representative of the data shown in the related graphs.

Fig. 2d the pictures should be labeled appropriately to orient the reader. Similarly, labeling should be added to Fig. 4g.

After discussing data from fig. 5, the authors presented for the first time data shown in Fig. 3 (Page 10 lines 257-264)

Reviewer #2 (Remarks to the Author):

This work evaluates the metabolic phenotype of leptin-deficient animals where markers of ER stress and autophagy are pharmacologically and genetically manipulated. The results show that at postnatal day 10, ob/ob mice show high levels of markers of ER stress and autophagy. Moreover, treatment of ob/ob mice with TUDCA ameliorates obesity-associated disorders while the lack of ATG7 in POMC neurons of ob/ob mice exacerbates obesity and glucose intolerance.

The study is elegantly performed, the results are solid and clearly support the conclusions. However, there are some major concerns:

1. As the authors mention throughout the text, the role of ER stress and autophagy in POMC neurons has been previously studied in several papers. Although it is true that this has not been done in leptin-deficient mice, the novelty of the study is somehow limited and this should be noted.
2. Related to the previous point, it has been published that the induction of Xbp1s in POMC neurons protects against diet-induced obesity and improves leptin and insulin sensitivity (ref 10). However, in the present study, XBP1s are increased in POMC neurons of ob/ob mice before they develop obesity and glucose intolerance. How do the authors reconcile this apparent discrepancy?
3. Also, the treatment with TUDCA (amelioration of ER stress) reverses obesity in ob/ob mice, which is the opposite to what XBP1s does in POMC neurons. This would mean that TUDCA acts in a different neuronal population or that TUDCA would act in POMC neurons but via XBP1s-independent pathways.
4. In figure 1 the levels of ER stress markers in POMC neurons are shown but do the authors know what is occurring in AgRP neurons? It may be possible that part of the effects of TUDCA found in figure 2 could be mediated by this neuronal population.
5. Figure 2. It is clear that TUDCA decreases body weight because its anorexigenic action, but it is surprising that energy expenditure is increased in ob/ob mice. Is this because a diet-thermogenic effect? I would recommend to represent energy expenditure without correcting by lean mass and using ANCOVA.
6. Figure 5. It would be interesting to show the phenotype of POMC-Cre, ATG7loxP/loxP mice.

We thank the Reviewers and the Associate Editor for their thoughtful treatment of our manuscript and their constructive comments. The manuscript has been extensively revised according to these comments, including additional new data. Revisions are highlighted in yellow.

Response to Reviewer 1's Comments

We thank Reviewer 1 who found our study interesting and novel. She/he also raised several comments:

General Comment. *"The study described in the manuscript is very hard to follow. The sequence of the data and how they are discussed in relation to the figures is very confusing. Some of the data shown are either not discussed or in discrepancy with other data shown. Picture should be properly labelled and the overall discussion of the data should be more in-depth"*

Response. We apologize if the initial manuscript was hard to follow and appreciate the reviewer's suggestion to improve the overall clarity of our manuscript. As suggested, the sequence in which the data are presented has been improved. In addition, all data are now mentioned and we corrected discrepancies between Figures and text in the Results section. In addition, images are now properly labeled, and we now provide a more in-depth discussion (see revised Results section and Figures and specific revisions below)

Specific Comment 1. "Lines 125-126 on page 5 authors stated: "During adulthood leptin deficiency only cause an increase in Atf6 and Xbp1 mRNA expression in the ARH (Fig.1f). However, Bip also is shown in the figure to be significantly altered while Atf4 seems to be significantly reduced in ob/ob mice. They are not mentioned and discussed."

Response. We thank the reviewer for pointing out this omission. We checked our data and only *Bip* and *Xbp1* mRNAs were increased in the ARH of adult ob/ob mice. We revised the Results accordingly (*Results*, p. 5, 1st paragraph).

Specific Comment 2. *"In Fig. 1j data on WT and ob/ob mice are a duplication of Fig. 1c."*

Response. We agree with the Reviewer that Fig. 1c was redundant because these data were also shown in Figure 1j. We therefore removed Figure 1c. These results are now only shown in one Figure (now Figure 1c).

Specific Comment 3. *"In Line 143 authors jumped from discussing data shown in Fig. 1k to data shown in Fig.4a. It is unclear why the authors organized the figures in a such confusing manner."*

Response. We apologize if this Figure sequence was confusing to the Reviewer. We have now pooled data shown in Figure 1k with data shown in Figure 4a (now Figure 1i).

Specific Comment 4. *"Fig 2a and 2b should be reconciled. It is known that ob/ob mice at birth and until about 6 weeks of age are indistinguishable from the WT control. In agreement with this, in figure 2b the authors show that post-weaning, ob/ob and WT mice have similar body weight from 3 weeks (at weaning) and up to 5 weeks of age."*

Response 5. We agree with the Reviewer that it is already known that body weights of *ob/ob* mice are similar to those of WT animals until after weaning. As suggested, we have now reconciled Figures 2a and 2b (now Figure 2a).

Specific Comment 5. *“Fig. 2 d show body composition done at 4 weeks of age. However, in the text (line 166) the authors referred to that as done at 6 weeks. Moreover, they stated in the text (line 167-168) that this reduction was not observed in 10 weeks old mice and referred to Fig.2d which, again, show data from 4 weeks old mice.”*

Response. We apologize for the confusion. Body composition shown in Figure 2d was measured at 4 weeks of age. Text in the Results section were revised accordingly (*Results*, p. 7, 1st paragraph). In addition, we forgot to include graphs showing body composition at 10 weeks of age, which is now shown in the revised manuscript (new Figure 2c).

Specific Comment 6. *“Fig. 2j reports ITT of 8 weeks old mice. However, in the legend the author reported that mice were 9 weeks old. It is unclear at what age insulin in fig 2k was measured and in what conditions (fed vs fasting).”*

Response. We apologize for this oversight and thank the Reviewer for pointing it out. ITT were performed at 9 weeks of age. Insulin levels were measured in fed animals at 10-12 weeks of age. This information has been included in the revised Figure and Figure legend.

Specific Comment 7. *“TUDCA-treated mice showed improved glucose tolerance test (Fig. 2h and i) but they show insulin resistant (Fig. 2j). No comments or discussion of these data is attempted by the authors.”*

Response. Although the effect of TUDCA treatment on ITT is not as marked as the effect it has on GTT, TUDCA-treated *ob/ob* mice did show improved insulin tolerance compared to vehicle-treated *ob/ob* mice. As suggested, the data are also discussed (*Discussion*, p. 11, 1st paragraph).

Specific Comment 8. *“Fig. 3c the outlined of the islet is off in a panel and in general the pictures do not seem to be representative of the data shown in the related graphs.”*

Response. We thank the Reviewer for raising this comment. Accordingly, we redrew the outline of the islets in Fig. 3c (now Figure 4) and now provide new images that are more representative of the data shown in the related graphs (now Figure 4).

Specific Comment 9. *“Fig. 2d the pictures should be labeled appropriately to orient the reader. Similarly, labeling should be added to Fig. 4g.”*

Response. We thank the Reviewer for raising this comment that will improve clarity of our paper. We have now added additional labeling in Figure 3d (now Figure 6a) and in Figure 4g (now Figure 5f).

Specific Comment 10. *“After discussing data from fig. 5, the authors presented for the first time data shown in Fig. 3 (Page 10 lines 257-264).”*

Response. Figure 3 is now presented prior to Figure 5 (*Results*, p. 8, 1st paragraph for Figure 3; p. 10, 1st paragraph for Figure 5 –now Figure 6).

Response to Reviewer 2's Comments

We thank the reviewer who found our study elegantly performed and that our results are solid and clearly support the conclusions. Several additional comments were made:

Comment 1. *“As the authors mention throughout the text, the role of ER stress and autophagy in POMC neurons has been previously studied in several papers. Although it*

is true that this has not been done in leptin-deficient mice, the novelty of the study is somehow limited and this should be noted.”

Response. Despite previous publications examining the importance of ER stress and autophagy in POMC neurons, we still feel that our paper is novel for at least two reasons. *First*, this is the first paper reporting a role for early life ER stress in lifelong metabolic regulation and hypothalamic development. *Second*, our paper represents the first link between leptin, autophagy and metabolic regulation and hypothalamic plasticity.

Comment 2. *“Related to the previous point, it has been published that the induction of Xbp1s in POMC neurons protects against diet-induced obesity and improves leptin and insulin sensitivity (ref 10). However, in the present study, XBP1s are increased in POMC neurons of ob/ob mice before they develop obesity and glucose intolerance. How do the authors reconcile this apparent discrepancy?”*

Response. We agree that our findings showing that elevated Xbp1 causes metabolic diseases can be paradoxical considering previous studies from Drs. Williams and Elmquist's labs reporting that genetic induction of Xbp1s in POMC neurons protects against DIO and improves leptin and insulin sensitivity (Cell Metab, 20(3):471-482, 2014). However, there are a few points to consider. *First*, Williams et al. examined the effect of transgenic overexpression of Xbp1s specifically in POMC neurons. It is important to note that these manipulations are restricted to one cell type and these effects could be canceled out or oppositely regulated with the recruitment of more cell populations, such as when animals are injected with chemical chaperones. With the TUDCA injections performed in our study, multiple neuronal populations are recruited including neurons that may override the effect of other neurons. In fact, recent data from Dr. Sternson's lab (Henry et al., eLife, 4:e09800, 2015) using single cell transcriptomics, revealed that Xbp1s is oppositely regulated in AgRP neurons. *Second*, Williams and colleagues examined selective overexpression of a single UPR factor (*i.e.*, Xbp1). However, in disease states (*e.g.*, in conditions of leptin deficiency) these factors are not commonly activated alone and multiple ER stress pathways are upregulated. For example, we found that *Atf4*, *Atf6*, *Xbp1*, *Bip*, and *Chop* are upregulated in the arcuate nucleus of *ob/ob* neonates. Therefore, conditional genetic models of are likely different than animal models of obesity. Moreover, Ozcan and colleagues (Cell Metab, 9(1):35-51, 2009) reported that mice genetically deficient in *Xbp1* in the CNS displayed higher levels of other ER stress pathways suggesting that compensatory mechanisms may occur in mice with specific deletion of a ER stress pathway.

These points are now discussed in the revised manuscript (Discussion, p. 13, 2nd paragraph).

Comment 3. *“Also, the treatment with TUDCA (amelioration of ER stress) reverses obesity in ob/ob mice, which is the opposite to what XBP1s does in POMC neurons. This would mean that TUDCA acts in a different neuronal population or that TUDCA would act in POMC neurons but via XBP1s-independent pathways.”*

Response. Our data are in agreement with previous studies from Dr. Ozcan's lab showing that administration of 4-Phenyl butyric acid (another chemical chaperone that relieves ER stress) in adult *ob/ob* mice ameliorates metabolic outcomes, including glucose homeostasis (Ozcan et al., Science, 313(5790): 1137-1140, 2006). As discussed above, this apparent discrepancy with Williams' study (Cell Metab, 20(3):471-482, 2014) could be explained by the fact that our TUDCA treatment acts on multiple ER stress pathways in several neuronal populations that could have opposite roles in energy balance regulation whereas Drs. Williams and Elmquist studied the role of Xbp1s alone

specifically in POMC neurons. This point is now discussed in the revised manuscript (Discussion, p. 13, 2nd paragraph).

Comment 4. *“In figure 1 the levels of ER stress markers in POMC neurons are shown but do the authors know what is occurring in AgRP neurons? It may be possible that part of the effects of TUDCA found in figure 2 could be mediated by this neuronal population.”*

Response. We thank the Reviewer for raising this interesting point. In response to this comment, we measured levels of ER stress markers in both POMC and AgRP neurons and found that *Atf4*, *Atf6*, *Xbp1*, and *Bip* are similarly increased in these two neuronal populations. However, because we only found an effect of TUDCA on POMC neuronal projections and not in AgRP/NPY circuits, it suggests that the effects of TUDCA are primarily restricted to POMC neurons. These new data are now shown in Figure 1d, and are mentioned in Results (p. 5, 1st paragraph). and Discussion (p. 12, 2nd paragraph).

Comment 5. *“Figure 2. It is clear that TUDCA decreases body weight because its anorexigenic action, but it is surprising that energy expenditure is increased in ob/ob mice. Is this because a diet-thermogenic effect? I would recommend to represent energy expenditure without correcting by lean mass and using ANOVA.”*

Response. We thank the reviewers for suggesting to represent energy expenditure without correcting by lean mass and using ANOVA. Using this approach, energy expenditure was not different between the various groups. A revised graph has been included in Figure 2 (now Figure 2e).

Comment 6. *Figure 5. It would be interesting to show the phenotype of POMC-Cre, ATG7loxP/loxP mice”*

Response. We thank the reviewers for this interesting suggestion. However, the metabolic and neurodevelopmental phenotype of *Pomc-Cre*, *Atg7*^{loxP/loxP} mice has already been published in our previous paper published in Cell Metabolism (Coupe et al, Cell Metabolism, 15:247–255, 2012). These published data are now discussed (p. 13, last paragraph)

We appreciate the constructive evaluation and inputs provided by the 2 reviewers and the Associate Editor. We feel that these changes extend the work and improve the overall impact and clarity of our manuscript.

REVIEWERS' COMMENTS:

Reviewer #1 (Remarks to the Author):

The authors responded adequately to previous critiques significantly improving the manuscript which is now suitable for publication. No further concerns.

Reviewer #2 (Remarks to the Author):

In response to comment 1, I admit that this is the first paper reporting a role for early life ER stress in lifelong metabolic regulation. However, the role of autophagy in POMC neurons as mediator of leptin response has been described before in adults (PMID: 22334718, cited in the manuscript).

On the other hand, I agree with the authors that TUDCA is likely acting on multiple ER stress several neuronal populations. However, the effects of TUDCA are primarily restricted to POMC neurons in the study (response to comment 4), which implies that TUDCA may use different hypothalamic routes in early versus adult stages?

In addition, in comment 2 they justify the current findings in part because Dr. Sternson's lab described that Xbp1s is oppositely regulated in AgRP neurons, but later on the authors affirm that in the present study Atf4, Atf6, Xbp1, and Bip are similarly increased in both POMC and AgRP.

We thank the Reviewers and the Associate Editor for their thoughtful treatment of our manuscript and their constructive comments. The manuscript has been revised according to these comments. Revisions can be seen with the track change feature.

Reviewer 1 had no further concerns.

Reviewer 2 had a few additional comments:

Comment 1. *“In response to comment 1, I admit that this is the first paper reporting a role for early life ER stress in lifelong metabolic regulation. However, the role of autophagy in POMC neurons as mediator of leptin response has been described before in adults (PMID: 22334718, cited in the manuscript).”*

Response 1. We thank the reviewer for this comment. The paper by Quan and colleagues (PMID: 22334718) showing that adult mice lacking Atg7 in POMC neurons display leptin resistance is indeed cited in our Manuscript (Discussion, p.13, last paragraph).

Comment 2. *“I agree with the authors that TUDCA is likely acting on multiple ER stress neuronal populations. However, the effects of TUDCA are primarily restricted to POMC neurons in the study (response to comment 4), which implies that TUDCA may use different hypothalamic routes in early versus adult stages?”*

Response 2. As pointed out by the Reviewer, our neuroanatomical results suggest that neonatal TUDCA treatment primarily acts on POMC neurons. Similarly, Cakir and colleagues (J Biol Chem, 288(24):7675–17688, 2013) showed that TUDCA injections in adult lean and DIO animals caused changes in aMSH levels but no changes in NPY peptide compared to vehicle-injected rats, supporting the idea that in adult animals TUDCA treatment also acts primarily on POMC neurons. This interesting point is now discussed in the revised manuscript (p. 12, last paragraph).

Comment 3. *“In comment 2 they justify the current findings in part because Dr. Sternson’s lab described that Xbp1s is oppositely regulated in AgRP neurons, but later on the authors affirm that in the present study Atf4, Atf6, Xbp1, and Bip are similarly increased in both POMC and AgRP.”*

Response 3. We agree that Dr. Sternson’s data appear to contrast with our findings. This discrepancy might be due to the fact that Dr. Sternson’s analysis was performed in adult animals whereas our study was done in neonatal mice. Although this is a potentially interesting issue to investigate, we feel that it is beyond the scope of the present study and Dr. Sternson’s reference was only used to provide answers to comment 2 of the Reviewer and is not included in our manuscript for clarity purpose.

We appreciate the constructive evaluation and inputs provided by the 2 reviewers and the Associate Editor. We feel that these changes extend the work and improve the overall impact and clarity of our manuscript.